# An analysis of farmers' experiences with deterrence methods and investment in mitigation of agricultural crop damage caused by geese

Sandie Lohse Sørensen[¤◉], Jesper Madsen[◉]*, Thorsten Johannes Skovbjerg Balsby

Department of Ecoscience, Aarhus University, Aarhus, Denmark

◉ These authors contributed equally to this work.
¤ Current address: Danish Hunters' Association, Rønde, Denmark
* jm@ecos.au.dk

## Abstract

The number of geese foraging in agricultural fields and causing damage to crops is increasing. Farmers attempt to reduce damage using passive, active, auditory, and combined deterrent measures, accommodation fields and, increasingly, derogation shooting. For protected geese like the barnacle goose *Branta leucopsis* and huntable species outside the hunting season, it is a legal requirement within the EU that other deterrent measures have proven insufficient before a derogation permit can be granted. However, there is a lack of guidance regarding the effectiveness of different measures. Via in-person interviews with 54 Danish farmers experiencing problems with wintering barnacle geese we analyse farmers' experiences with deterrence methods to provide an overview of their effectiveness, defined by duration and area coverage. The information obtained is far more extensive than what could realistically be achieved through scientific experiments. We check the validity of responses by comparing reports with existing scientific evidence. Passive deterrents (e.g., scarecrows) cover a few hectares and have a duration effect of 4–6 days, but only until the geese habituate. Active measures (e.g., a person walking into the field) and auditory deterrents (firing scare shots) have high area effect but short duration. Largest area/duration effects are achieved using gas cannons, sound deterrents and derogation shooting. Intensified active deterrence or increased density of passive deterrents enhance effectiveness but require greater investments of time and resources. Effective derogation shooting requires that hunters can respond quickly when needed. Hunting lease agreements regarding hunters' contributions to deterrence and derogation can enhance cooperation and problem-solving. In addition, cooperation between neighboring farmers, including accommodation areas, furthers effectiveness. The cost of geese (yield loss plus time/materials) can have a sizeable impact on the farmers' operation profits. Lack of effort may be due to farmers either coping with the problem, having given up deterrence, or unawareness of more effective deterrence.

**Data availability statement:** The survey protocols (in Danish) underlying the results presented in the study are available from Aarhus University / DCE Scientific Report: Sørensen, S. L., Balsby, T. J. S., & Madsen, J. (2025). Afværgemidler i forhold til gæs: Undersøgelse af landmænds erfaringer med traditionelle og innovative metoder. Aarhus University, DCE - Danish Centre for Environment and Energy. Videnskabelig rapport fra DCE - Nationalt Center for Miljø og Energi Nr. 649 https://dce.au.dk/fileadmin/dce.au.dk/Udgivelser/Videnskabelige_rapporter_600-699/SR649.pdf The data set used for analysis has been uploaded to Zenodo and is publicly available: Madsen, J., Sørensen, S. L., Balsby, T. J. S., & Mikkelsen, P. (2025). Data from: An analysis of farmers' experiences with deterrence methods and investment in mitigation of agricultural crop damage caused by geese (0.1.0) [Data set]. Zenodo. https://doi.org/10.5281/zenodo.17936702.

**Funding:** The project was primarily funded by a grant from the Danish Agency for Green Transition and Aquatic En-vironment to Jesper Madsen (www.sgav.dk)(Jagttegnsmidlerne 2023; ref. 120362), with support from a grant by the 15 June Foundation to Jesper Madsen (www.15junifonden.dk) (project Samforvaltning i praksis – forsøg med bramgæs i Guldborgsund Kommune; ref. 2022-022).

**Competing interests:** The authors have declared that no competing interests exist.

## Introduction

For several decades, goose populations have grown dramatically in many parts of the northern Hemisphere [1], causing increasing conflicts with agricultural interests with high socioeconomic costs [2,3]. Farmers have traditionally tried to mitigate the damage by various scaring techniques, however, with mixed and mostly frustrating results. Deterrence methods vary from classic scarecrows, flags and silhouette figures of dogs/wolfs to more advanced measures such as drones, sound systems, and derogation shooting [4]. Methods can be categorized as visual, auditory, lethal, combined techniques (exploiting the geese's innate fear of predators and/or hunting) and habitat modification [5], and they may be passive or active, manual or automatic. Managers and scientists have tested aspects of the effectiveness of different deterrence methods [e.g., 6–12]. These studies often provide good indications of scaring effects at a local level but are often incomplete as they typically test one or two methods in a specific situation, targeting a single goose species in a specific crop/habitat, and at a given time of the season. This makes it difficult to generalize from these studies to other species and situations, as well as to compare the effectiveness of different deterrents.

Among stakeholders, for example expressed through the European Goose Management Platform under AEWA/UNEP and nationally (e.g., Denmark) there is a call for better evidence-based guidance on effective deterrence methods. However, providing this is a huge task which is very difficult to achieve with a traditional scientific experimental approach. Here, we take an alternative approach. Instead of extending classical experiments with deterrents, this study explores the extensive practical know-how about the effectiveness of deterrents in relationship to geese accumulated by users (farmers and hunters) in Denmark over recent decades. Through in-person interviews we have collected and analysed their experiences, and the validity of the results are compared with existing knowledge based on scientific studies. We have also asked farmers about their costs, i.e., perceived yield loss caused by goose foraging, time spent, and equipment used to handle the goose problem, enlightening the scale of the problem and scope for additional investment in cost-effective deterrence efforts. Finally, we have asked about experiences with local collaboration among farmers and their rental hunters to tackle the problems on a wider scale, including creation of alternative foraging areas for geese where they do not cause damage. To ensure that our work will also provide input to the authorities' need for better documentation of the effectiveness of different deterrence methods, we have initially informed the authorities responsible for goose management in Denmark about the project objectives and discussed interview questions of relevance for their administrative purposes.

In Denmark, particularly increasing numbers of barnacle geese *Branta leucopsis* and greylag geese *Anser anser* pose the greatest challenges to agriculture. Approximately 270,000 barnacle geese winter in Denmark [13], often forming flocks with thousands of individuals. They primarily forage on winter-green fields, such as winter wheat, which can be vulnerable to goose grazing, particularly if it continues into spring [14]. Additionally, barnacle geese graze intensively on saltmarshes and

cultivated grasslands until mid-May before migrating to their breeding grounds, causing significant yield losses [15]. Grey-lag geese are less numerous during winter, but peak with up to 147,000 in summer in Denmark [16]. They do not gather in large foraging flocks and are mostly problematic during the summer months when they forage on unharvested cereal or cultivated grassland. While barnacle geese are protected under the EU Birds Directive and fully protected, greylag goose are huntable (from August 1 to January 31; they can also be shot under derogation in February and July). Differences in population size, flock sizes, foraging season, and huntability between barnacle and greylag geese mean that barnacle geese by far are perceived by farmers as the bigger problem. Therefore, this study focuses on the challenges associated with wintering barnacle geese.

According to the Danish wildlife damage regulations [17], barnacle geese can be shot under derogation under prior authorization if they cause damage to fields (from September 1–31 May). To obtain derogation permit for barnacle geese outside the period from September 1 to January 31 authorities require that non-lethal deterrence has been attempted. Therefore, an application for permission must specify whether deterrent measures have been used and which ones. According to the regulation, the Nature Agency may impose additional conditions, for example that deterrents have been deployed and proven insufficient and that deterrents recommended by the Nature Agency have been used and are still in use during derogation shooting [18]. Unlike most of the surrounding countries in northwest Europe, Danish farmers are not paid compensation for damage caused by geese, and there are no targeted subsidy schemes to provide peaceful accommodation areas for geese where they do not cause damage [4]. Scaring geese off from vulnerable fields, eventually using hunting or derogation shooting if other non-lethal methods have been exhausted, is therefore the main tool used by Danish farmers to mitigate the damage caused by geese.

Legally, the use of non-lethal deterrence is thus an essential requirement in managing the agricultural conflict with geese, especially barnacle geese, which are not huntable. However, knowledge about the behavioral and spatial responses to deterrence is limited. For farmers to implement the most effective deterrent measures, and for authorities to provide the best guidance and ensure compliance with national and international regulations, more knowledge is needed on how to deploy effective deterrent measures. The Nature Agency has developed a catalog of recommended wildlife deterrents for preventing significant damage and nuisance caused by wildlife [18]. This catalog describes recommended deterrents, but not all deterrents are included, and their effectiveness is not assessed.

For farmers the goal is to find the most cost-effective deterrents. We define 'effective' as a measure to deter geese over the largest possible area for the longest possible time, with minimal effort and costs (time consumption and price of device).

## Methods

The main content of the project consists of structured interviews with 54 users of goose deterrent measures with the aim of gathering experiences from the use of a wide range of deterrent methods. The experience-based synthesis is compared with existing literature on controlled trials of goose deterrent methods to assess whether there is consistency between the two types of knowledge. Prior to the interviews, an initial meeting (April 2024) was held with representatives from the Danish Nature Agency and the Agency for Green Transition and Aquatic Environment responsible for goose management in Denmark. The purpose of the meeting was to introduce the overall project objectives, discuss current practices for issuing derogation shooting permits for geese, and clarify the authorities' needs for specifications on the use of deterrence measures prior to issuing permits. Additionally, we aimed to specify the workflow and knowledge needs to be addressed by the interviews. The meeting outcomes contributed to the formulation of the interview protocol.

### Interviews and survey protocol

Structured interviews were conducted as survey-based studies [19] to map users' experiences with deterrence methods, enabling an overview of their assessments of the practical effectiveness, time consumption, and economic costs caused

by geese and use of deterrence measures. The survey was also designed to uncover additional perspectives through reflective questions with open-text responses.

The electronic survey was developed and structured in Survey-Xact (Rambøll). The survey protocol was designed with six main themes: 1) respondents' agricultural background, 2) hunting interests, 3) challenges with geese and economic losses, 4) experiences with deterrence measures, 5) experiences with alternative feeding areas, and 6) experiences with derogation shooting. The survey protocol is provided in Appendix 1 in Sørensen, Madsen and Balsby [20]. Key glossary explaining some of the terms used both in the survey protocol and in the data analysis is shown in Table 1.

We included all types of classic deterrence methods described in the Nature Agency catalog [18] but also aimed to cover as wide a range as possible to gather information about more innovative methods such as the use of bird-alerts (scare sounds), drones, lasers, and 'hanging human figures'. To ensure quality and fine-tune the survey protocol, two pilot interviews with experienced users were conducted.

The survey interviews were conducted in person, where the interviewer (exclusively SLS) read the questions aloud and recorded the respondent's answers directly on the digital platform. The respondent was able to read the questions and view the answer options during the interview. This method allowed for a series of detailed questions to be asked under each theme, as the respondent was required to complete the survey protocol. Each interview took between 50–120 minutes and was conducted on the respondent's property. All interviews took place from May to August 2024. Most of the survey questions were designed with answer options, but there were also selected questions with the possibility for open-text responses, which allowed for more nuanced answers.

The respondents were selected based on the following criteria: The respondent had to 1) be a farmer or employee in agriculture, 2) have challenges with geese or work for farmers with goose-related issues and, 3) have experience with deterrence methods aimed at geese. As the Nature Agency has contacts with citizens locally across the country, their wildlife consultants were asked to propose possible respondents in their respective districts. To achieve a larger number of respondents, snowball sampling was used. This method relies on initially contacted respondents referring to other

**Table 1. Terms used in interviews and analysis of deterrence effects and economics related to goose damage and scale of problem.**

| Variable | Explanation | Categorisation |
|---|---|---|
| Degree of burden | The degree of burden that respondents experience from foraging geese on their fields | 1 = no burden, 2 = small burden, 3 = moderate burden, 4 = large burden, and 5 = overwhelmingly large burden |
| Duration effect | The time (in intervals of days) deterrent measure can keep geese away with a single effort or from the time it is initially deployed. | 1: < 1 day; 2: 1–3 days; 3: 4–7 days; 4: 7–21 days; 5: > 21 days |
| Area effect | The area (in intervals of hectares) from which a deterrent measure or a single effort can keep geese away shortly after deployment | 1: < 1 ha; 2: 1 ha; 3: 2–5 ha; 4: 6–10 ha; 5: 11–20 ha |
| Density of deterrents | The density of passive deterrents put up in each field exposed to goose grazing | Intervals of densities in hectares |
| Effectiveness of deterrence methods | How well a deterrent measure performs overall in achieving long duration effect and large area effect. | The combination of duration and area effects (time x ha) |
| Resource costs | Costs of deterrence equipment and fuel costs (EURO per year) and hours spent on scaring efforts converted to EURO per week. | Intervals of EURO per year or week |
| Economic loss | The respondents' estimated economic loss (EURO per year) due to yield loss caused by goose foraging on crops, despite mitigation efforts. | Intervals of EURO per year |

potential respondents within the criteria [19]. This allowed the survey to be conducted with 59 respondents, of whom 54 considered barnacle geese their greatest challenge, while only five reported greylag geese as their biggest issue. As a result, we were only able to statistically assess the data on barnacle geese. The 54 respondents were distributed in four geographical areas: Jutland (n = 8), Fyn, Langeland and Tåsinge (n = 14), Zealand (n = 14), and Lolland-Falster (n = 18). Due to another ongoing goose management study in Guldborgsund Municipality (Lolland-Falster), we had contact to farmers in that area, which were used for this study as well. This resulted in an overrepresentation of respondents from that area.

Informed consent was obtained from all participants prior to data collection. All respondents were initially contacted by phone, where the background and purpose of the study were explained, the respondent criteria were verified, and the respondent was given the choice to participate based on this information. Upon positive verbal confirmation of participation, a physical meeting was scheduled for the respondent's 'home ground'. As an introduction to the physical meeting, respondents were provided with clear information regarding the purpose of the study, methodology, data handling procedures, and assurance of anonymity. Participation was voluntary, and respondents were informed of their right to withdraw at any time and were given the opportunity to ask questions throughout the process. Consent was documented verbally both before and during the interview and in writing via the questionnaire. No sensitive personal data was collected, and all identifying information (names and email addresses) was anonymized in compliance with GDPR requirements. The study did not require approval from a research ethics committee, as it did not involve vulnerable groups, health data, or sensitive topics.

## Statistical analyses

For the statistical analysis of interviews, data from Survey-Xact was organized into a relational database (MS Access).

**Duration and area effects.** By coding the duration and area effects of deterrence methods (Table 1), both variables followed a multinomial distribution. We therefore analysed each variable in relation to the type of deterrence method using a generalized mixed model with a multinomial distribution. Additionally, the model for duration effect also included the degree of burden, based on the hypothesis that the higher the burden (and thus the higher the goose density), the faster the geese would return to a field. Respondent was included as a random factor in all models on effect, because respondents reported on multiple deterrence methods.

Thus, the model for duration effect was:

$$\text{Duration effect} = \text{deterrence method} + \text{burden}$$

The model for area effect was:

$$\text{Area effect} = \text{deterrence method}$$

For passive deterrence methods, the density at which they were placed can affect the duration effect. We tested this relationship with the following model:

$$\text{Duration effect} = \text{density} + \text{deterrence method}.$$

**Economic aspects.** We asked respondents to give an assessment of the burden and costs of geese on an annual basis. Questions related to (i) an overall qualitative assessment of the burden of geese in relation to the season of goose grazing and the area of farmland under cultivation, (ii) the costs in terms of perceived estimated yield loss caused by the geese and (iii) the resource costs in terms of purchase of deterrence equipment, driving costs (fuel costs) and time spent on mitigation actions. To convert labour hours to monetary costs, we used a standard rate for an agricultural assistant in Denmark of 24.0 EURO per hour [21].

To test if the burden of goose grazing (1–5) varied between seasons of goose grazing and the amount of area under cultivation we used a generalized linear mixed model with multinomial distribution:

$$Burden \ = \ season \ + \ area$$

We conducted similar analysis regarding resource costs and estimated economic yield loss. In the analyses, we only included reports that indicated either autumn, winter, or spring.

We tested the relation between the perceived economic yield loss and the burden of goose grazing. Farmers assessed the economic loss on an interval scale (ranked scale). The economic loss thus followed a multinomial distribution, and the model describes the probability of an increase in economic loss with an increase in burden.

To test whether the perceived economic yield loss motivated farmers to invest more in mitigation, we analyzed the relationship between resource cost and the perceived economic yield loss. The farmers' investment was quantified as the resource costs, i.e., cost of deterrence equipment, fuel costs and salary cost for one week of mitigation. This means that resource cost had many possible outcomes, and therefore the resource cost was divided into four groups: <670, 670–1340, 1340–2680, or >2680 EURO per year. The total resource costs, however, depend on the length of the mitigation period. Since we did not know for how long time respondents mitigated per year, two scenarios were chosen. We calculated the total costs based on 1 and 20 weeks of mitigation which was known to reflect the spectrum of mitigation periods in the study in Guldborgsund Municipality (J. Madsen unpublished data). We assumed that the costs for deterrent equipment and fuel were constant because this was reported per year by the respondents. Hence, only labour time was affected by the mitigation period.

Salary cost and deterrence equipment cost followed a multinomial distribution. These models thus modeled the probability of an increase in cost as a function of the economic yield loss. Since the costs for equipment were not given in intervals, the total resource costs were treated as continuous variables. To model the total costs, we assumed that they followed a Poisson distribution. We tested the pairwise comparisons using multiple contrast.

### Effects of collaborative agreements

We analyzed the relationship between respondents' hunting lease agreements with hunters, where derogation shooting either was or was not part of the agreement. This was compared with respondents' satisfaction with the hunters' preparedness to take action when needed. We tested this with a $\chi^2$ test.

We used proc glimmix or proc genmod in SAS 9.4 [22] to perform the analyses.

## Results

The respondents have experience with many different deterrence methods, ranging from experience with a single to 15 different methods. Therefore, in the following section, data will be presented with differences in sample sizes between tested deterrence methods.

### Crops and seasons affected by geese

Out of the 54 respondents, 74% rate winter wheat as the crop most affected by barnacle geese. The remaining 26% are divided between seed grass, grass used for grazing or haycutting, winter oilseed rape, spring crops, vegetables or unknown. Forty five percent of the respondents rate spring as the season when geese cause the most problems, and 7% indicate that it is during the entire wintering period that barnacle geese are an issue.

### Duration effects of deterrence methods

The deterrent methods used by the respondents are grouped into passive, auditory, active, and combined techniques. Combined techniques are methods that use both visual and auditory deterrence, functioning as both passive and active

measures and include inflatable howlers, dead geese in the field, and drones. Leaving dead geese in the field after derogation shooting is a method that 24 respondents have experience with. The idea is that by placing dead geese in the field, one attracts white-tailed eagles *Haliaeetus albicilla*, for which the barnacle geese have a natural fear, thereby scaring them away from the field.

Active deterrence methods have the shortest median duration effect of the methods used (Fig 1). Among the active methods, only 'handheld lasers' have a median duration of more than one day (though there is limited information), while 'walking person,' 'car/ATV – driving into the field,' and 'dog off-leash' are all assessed to have a median duration of less than one day. Auditory deterrence methods are among those with the longest median duration effect, with 'scare sounds' and 'gas cannons' having the highest values of 7–21 days. 'Derogation shooting' is also considered to have a relatively high duration effect, with a median value of 7–21 days. It is indicated that 'hanging human figure' has a high median duration of effect of over 21 days, but only three responses were received.

The degree of burden shows no relationship with the duration effect (generalized linear mixed model, $F_{2,263} = 0.19$, $p = 0.456$). The duration effect is significantly different between the 11 types of deterrence methods (generalized linear mixed model, $F_{10,263} = 13.4$, $p < 0.001$). The relative parameter estimates for fixed effects provide a ranking of the various types of methods (Table 2). The parameter estimates are relative because they are all given in relation to 'agricultural equipment,' which is set to zero, where negative estimates indicate shorter duration and positive estimates longer duration of the method. The parameter estimates indicate that 'gas cannon' is the most effective method, followed by 'agricultural equipment' and 'derogation shooting'. Methods such as 'walking person,' scare shots', 'driving with car/ATV', and 'dog

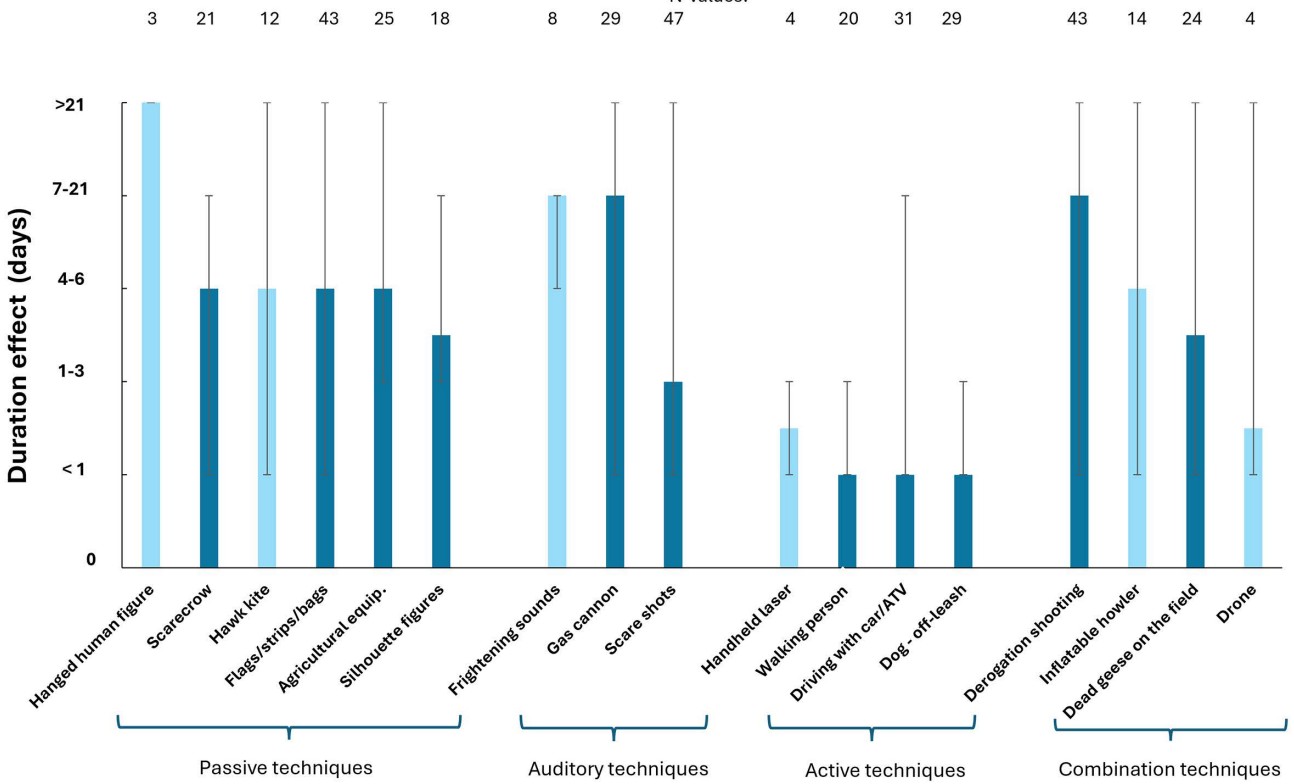

**Fig 1. The median duration effects of different deterrence methods in day intervals.** Vertical lines show minimum/maximum. Dark blue bars indicate the methods included in the statistical pairwise model, as opposed to light blue bars where the sample size does not allow them to be included.

**Table 2. Pairwise tests of the duration effects of deterrence methods with indication of p-values.**

| | Dead geese in field | Scare-crow | Walking person | Gas cannon | Derogation shooting | Silhouette figure | Scare shots | Driving with car/ATV | Flags/-strips/-bags | Dog off-leash |
|---|---|---|---|---|---|---|---|---|---|---|
| **Dead geese in field (−0,80)** | | | | | | | | | | |
| **Scarecrow (−0,86)** | 0.917 | | | | | | | | | |
| **Walking person (−4,58)** | <.001 | <.001 | | | | | | | | |
| **Gas cannon (0,65)** | 0.006 | 0.006 | <.001 | | | | | | | |
| **Derogation (−0,19)** | 0.198 | 0.177 | <.001 | 0.059 | | | | | | |
| **Silhouette figure (−0,91)** | 0.848 | 0.929 | <.001 | 0.006 | 0.169 | | | | | |
| **Scare shots (−2,27)** | 0.002 | 0.005 | <.001 | <.001 | <.001 | 0.01 | | | | |
| **Driving with car/ATV (−3,53)** | <.001 | <.001 | 0.131 | <.001 | <.001 | <.001 | 0.008 | | | |
| **Flags/strips/bags (−0,73)** | 0.881 | 0.795 | <.001 | 0.002 | 0.174 | 0.726 | 0.001 | <.001 | | |
| **Dog off-leash (−4,57)** | <.001 | <.001 | 0.989 | <.001 | <.001 | <.001 | <.001 | 0.101 | <.001 | |
| **Agriculture equip. (0)** | 0.1318 | 0.119 | <.001 | 0.199 | 0.675 | 0.114 | <.001 | <.001 | 0.116 | <.001 |

Blue marked cells indicate a significant difference, where methods in the top row have a significantly longer duration than the methods in the column, for example, 'dead geese in the field' has a significantly longer duration effect than 'walking person'. Conversely, orange marked cells show that the method in the left column has a significantly longer duration than the method in the row, for example, 'gas cannon' has a significantly longer duration effect than 'dead geese in the field'. Values in parentheses indicate the relative parameter values set in relation to 'agricultural equipment' put out in the field (set to 0), where positive values have longer duration effects and negative values have shorter duration effects than 'agricultural equipment'. The tests are done as multiple contrast. See Appendix 2 in Sørensen, Balsby and Madsen [20] for details.

off-leash' have the shortest duration effect. A group consisting of 'dead geese in the field', 'scarecrows', and 'flags/strips/bags' do not differ significantly from one another.

The results also show that derogation shooting is considered to have a significantly longer duration effect than both 'dog off-leash', 'driving car/ATV', 'walking person', and 'scare shots'. The four passive methods included in the statistical analysis, namely 'scarecrow', 'flags/strips/bags', 'silhouette figure', and 'agricultural equipment' have a significantly longer duration effect than all the active methods. Additionally, the passive methods have a significantly longer duration effect than 'scare shots', which belongs to the auditory category of deterrence methods (Table 2).

### Effects of density of passive deterrence on duration effect

To analyse if there is an effect of the density of passive deterrence methods on duration effect, we group the three most used passive deterrence methods scarecrows, flags/strips/bags and silhouette figures, which do not show a significant difference in duration effect (generalized linear mixed model, $F_{2,20} = 0.028$, $p = 0.762$). An analysis of the density of these three passive methods grouped shows a significant effect on the duration effect (generalized linear mixed model, $F_{1,20} = 4.86$, $p = 0.007$). To illustrate the duration effect for different densities of passive deterrents, we set the density category 'one per 10-20 ha' to a value of zero. This makes it possible to compare the relative difference to the other four

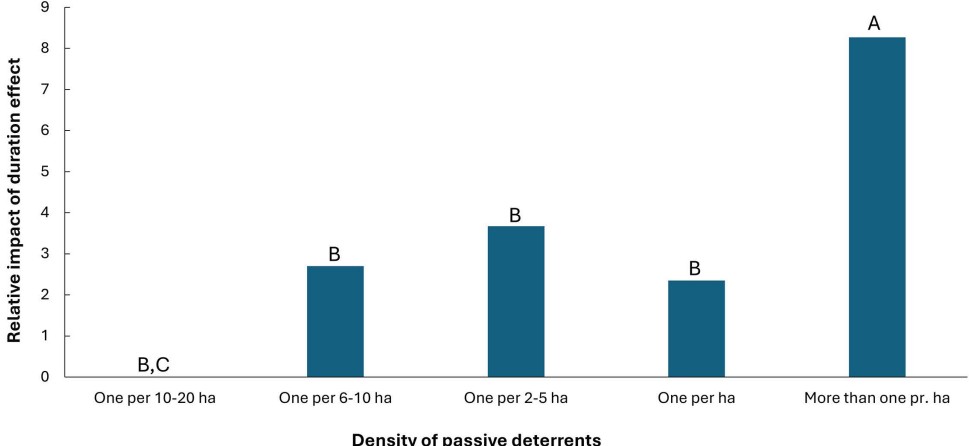

**Fig 2. Relative duration effect of each of increasing densities of passive deterrence methods set relative to 'one per 10-20 ha' (set to 0).** Letters indicate significant differences, meaning that 'more than one per ha' is significantly different from all others (A against B) (p < 0.01 in all pairwise posthoc tests), and 'one per 2-5 ha' is significantly different from 'one per 10-20 ha' (C) (p = 0.046 in pairwise posthoc test).

categories (one deterrent per: 6–10 ha, 2–5 ha, one ha, and more than one per ha)(Fig 2). As shown, respondents who make a greater effort to place passive deterrents closer than one per ha obtain a significantly longer duration effect than the other densities of passive deterrents. Additionally, the category 'one deterrent per 2-5 ha' has a significantly longer duration effect than 'one deterrent per 10-20 ha'.

## Area effects of deterrence methods

The active deterrence methods have the highest median area coverage effect, 11–20 ha and >20 ha, followed by auditory methods, and passive methods have the lowest effect (1–10 ha) (Fig 3).

The area effect varies significantly between the ten types of deterrents tested ($F_{9, 216}$ = 18.16, p < 0.001). The results of the statistical pairwise comparisons show that 'driving with car/ATV" and 'scare shots' have significantly higher area effects than all other tested deterrents (Table 3). Handheld lasers and drones are indicated to have large area effects (>21 ha), but unfortunately, there are too few samples for them to be included in the pairwise statistical analysis.

## Combined area and duration effects

To express the effectiveness of the various deterrents in time and space, we compare the median values of both area and duration effects and categorize them into four quadrants of the outcome space: both low duration and area effect; high duration and low area effect; low duration and high area effect; both high duration and area effect (Fig 4). It should be noted that we have included all reported deterrents, even if for some we have fewer than 10 data points. This presentation does not take into account that geese may habituate to certain deterrents over time, especially stationary ones (passive, auditory, and combined). In the highest quadrant, derogation shooting, gas cannons and frightening sounds are found, while in the lowest quadrant, the passive deterrence are found, except for 'hanging human figure' which appears to have a higher duration effect that other passive methods. Active methods score low on duration but high on area effect.

## Economic costs

In response to the question about the extent to which foraging geese are perceived as a burden on their fields, 65% of respondents indicate that it is a moderate burden, 22% that it is a significant burden, and 13% that it is an overwhelming burden. We

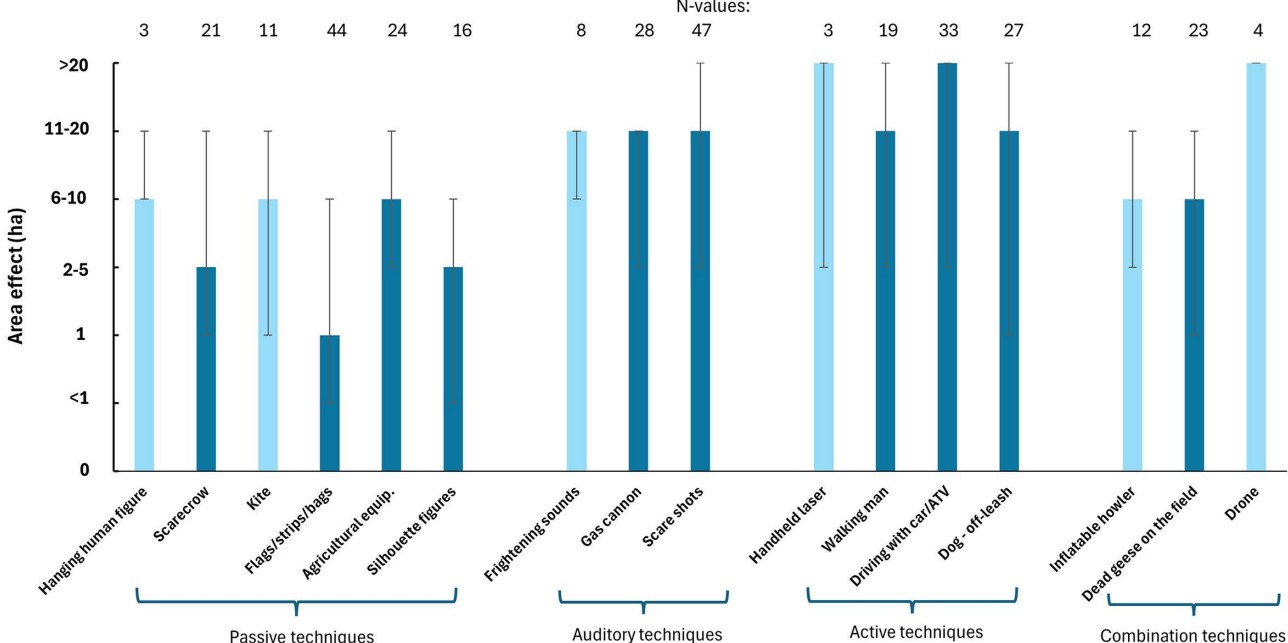

**Fig 3. The median area effect of deterrence methods in intervals of hectares.** Vertical lines show minimum/maximum. Dark blue bars indicate the methods included in the statistical pairwise model, as opposed to light blue bars where the sample size does not allow them to be included.

analyse whether the degree of burden is related to the season identified by respondents as the most problematic due to geese and the area under cultivation, and we find no significant effects (Table 4). Resource costs used for handling geese are significantly related to the area under cultivation. Thus, respondents with larger areas under cultivation tend to experience higher resource costs for deterrence. The economic yield loss is not related to season nor area under cultivation (Table 4). Furthermore, the degree of burden shows a significant relation with the reported economic yield loss ($X^2_3 = 3.32$, p = 0.190).

The perceived economic yield loss by farmers shows a significant positive relationship with the reported labour costs spent on scaring ($X^2_3 = 10.41$, p = 0.015): Similarly, there is a positive relationship with the costs for deterrence equipment and fuel for driving ($X^2_3 = 9.47$, p = 0.024, Fig 5). Fig 5 shows a large variation in costs within each category of reported economic loss. Farmers in the low-loss category are less likely to invest large sums for deterrent equipment, however, they spend relatively much time on scaring activities. Conversely, in the highest loss category (>13,400 EURO), there are respondents that invest substantial amounts in deterrent equipment, but also respondents who invest very little in terms of equipment and time. Logically, the costs will increase with the time of goose presence. Hence, scenarios of the length of the period of mitigation suggest that on average the resource costs increase by a factor of 1.13 per week and by a factor of 3.4 going from a situation with one week to 20 weeks of mitigation (Fig 5). Again, some farmers appear to invest little time and equipment despite reporting high levels of yield loss.

### Derogation shooting of geese

Of the 54 respondents surveyed, 47 (87%) have hunters who carry out derogation shooting of barnacle geese on their behalf. Among the 47 respondents, 28 (60%) have entered into a hunting lease agreement with the hunters. Of these, 13 (46%) have entered into an agreement that includes specific terms requiring the hunters to perform goose derogation shooting. Of the 47 respondents with hunters conducting derogation shooting, 56% report that hunters 'often' come to perform derogation when needed. The remaining 44% are evenly split between 'not always' and 'rarely,' while 2% respond

Table 3. Pairwise tests of the area effect of the deterrence methods with indication of p-values.

| | Dead geese in field | Scare-crow | Walking person | Gas cannon | Silhou- ette figure | Scare shots | Driving with car/ATV | Flags/-strips/- bags | Dog off-leash |
|---|---|---|---|---|---|---|---|---|---|
| **Dead geese in field (0,24)** | | | | | | | | | |
| **Scarecrow (−2,15)** | <.001 | | | | | | | | |
| **Walking person (0,89)** | 0.271 | <.001 | | | | | | | |
| **Gas cannon (0,60)** | 0.501 | <.001 | 0.612 | | | | | | |
| **Silhouette figure (−2,62)** | <.001 | 0.466 | <.001 | <.001 | | | | | |
| **Scare shots (2,42)** | <.001 | <.001 | 0.004 | <0.01 | <.001 | | | | |
| **Driving with car/ ATV (3,28)** | <.001 | <.001 | <.001 | <.001 | <.001 | 0.068 | | | |
| **Flags/strips/bags (−3,16)** | <.001 | 0.052 | <.001 | <.001 | 0.338 | <.001 | <.001 | | |
| **Dog off-leash (−1,17)** | 0.086 | <.001 | 0.620 | 0.266 | <.001 | 0.008 | 0.001 | <.001 | |
| **Agricultural equip. (0)** | 0.654 | <.001 | 0.129 | 0.248 | <.001 | <.001 | <.001 | <.001 | 0.029 |

Blue marked fields indicate a significant difference, where methods in the top row have significantly greater area effect than the methods in the column, for example, 'dead geese in the field' has a significantly higher effect than 'scarecrow'. Conversely, orange marked fields indicate that the method in the left column has significantly greater area effect than the method in the row, for example, 'walking person' has a significantly higher effect than 'scare-crow'. Values in parentheses indicate the relative parameter values set in relation to 'agricultural equipment' put out in the field (set to 0), where positive values have a greater area effect and negative values have a smaller area effect than 'agricultural equipment'. The pairwise comparisons are made as multiple contrast. See Appendix 2 in Sørensen, Balsby and Madsen [20] for details.

'never'. Respondents are significantly more satisfied with hunters when they have entered into a hunting lease agreement that specifically includes goose derogation as part of the agreement ($X_1^2 = 15.4$, $p < 0.01$) (Table 5).

## Accommodation fields

Among the 54 respondents, 29 use accommodation fields, and 20 of them use this effort as a deliberate attempt to lure barnacle geese away from more vulnerable crops during the wintering period. It should be noted that 15 out of the 18 respondents from Lolland-Falster use such accomodation fields. This group of respondents has been involved in the longer-term project concerning local goose management of barnacle geese, which includes the establishment of accommodation fields with sugar beet waste after harvesting [23]. Accommodation fields are also used in other geographical areas, such as maize waste fields, grass fields, and meadow areas. Nine respondents set aside more than 40 ha as accommodation, while 20 respondents set aside 40 ha or less.

When asked whether accommodation fields have a mitigating effect, 59% answer "yes," 24% answer "no," and 17% answer "don't know." Of the 29 respondents who use accommodation fields, 21 state that the accommodation fields are hunting-free, while eight report that the accommodation fields are not hunting-free.

Regarding the question of how well the respondents assess the effectiveness of their "overall deterrence activities," 78% of the 54 respondents answer "to some extent," 16% answer "satisfactory", while the remaining 6% are split between "almost not at all" and "not at all".

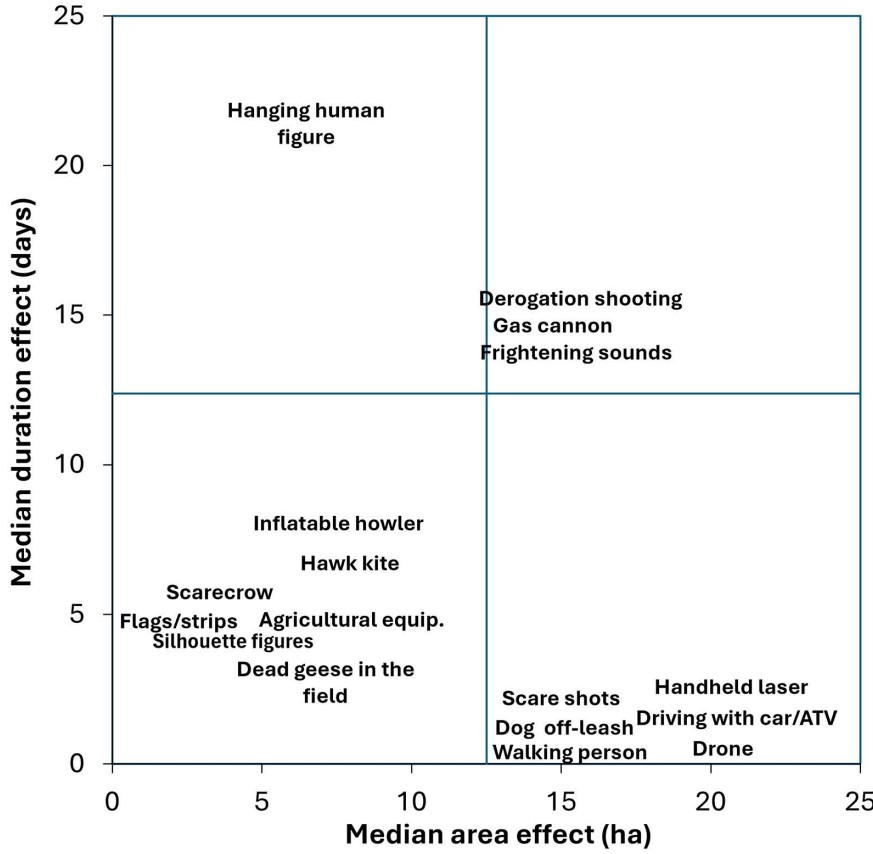

**Fig 4. The effectiveness of the deterrence methods used, expressed by the intersect between the median duration and area effects, divided into four quadrants.** For derogation shooting where no data is available to assess the area effect, it is assumed that the area effect corresponds to the median for scare shots. For durations that are estimated to be longer than 21 days, the value is set at 22. Likewise, for area coverages larger than 20 ha, the values are set at 21.

**Table 4. Test of the extent of the goose problem in relation to season with the highest goose problem and area in rotation.**

| | Degree of burden | | | Resource costs one week | | | Economic yield loss | | |
|---|---|---|---|---|---|---|---|---|---|
| Effect | Df | F | p | Df | F | p | Df | F | p |
| Season with highest problem | 2,44 | 1.11 | 0.340 | 2,32 | 2.57 | 0.092 | 2,33 | 0.97 | 0.391 |
| Area in rotation (ha) | 1,44 | 0.35 | 0.557 | 1,32 | 4.96 | 0.033 | 1,33 | 1.60 | 0.214 |

The extent of the problem is expressed as the perceived degree of burden with geese, estimated resource costs for one week of mitigation activity and the perceived economic yield loss caused by geese, respectively. Season with highest level of problems with goose damage is given in autumn, winter and spring.

## Discussion

The winter period, when there is no vegetation growth and foraging days are short, constitutes an energetic bottleneck to barnacle geese [24] but the modern agricultural land, with wintergreen fields and waste crops, provides the geese with improved opportunities to meet their energy needs during the winter period [25,26].

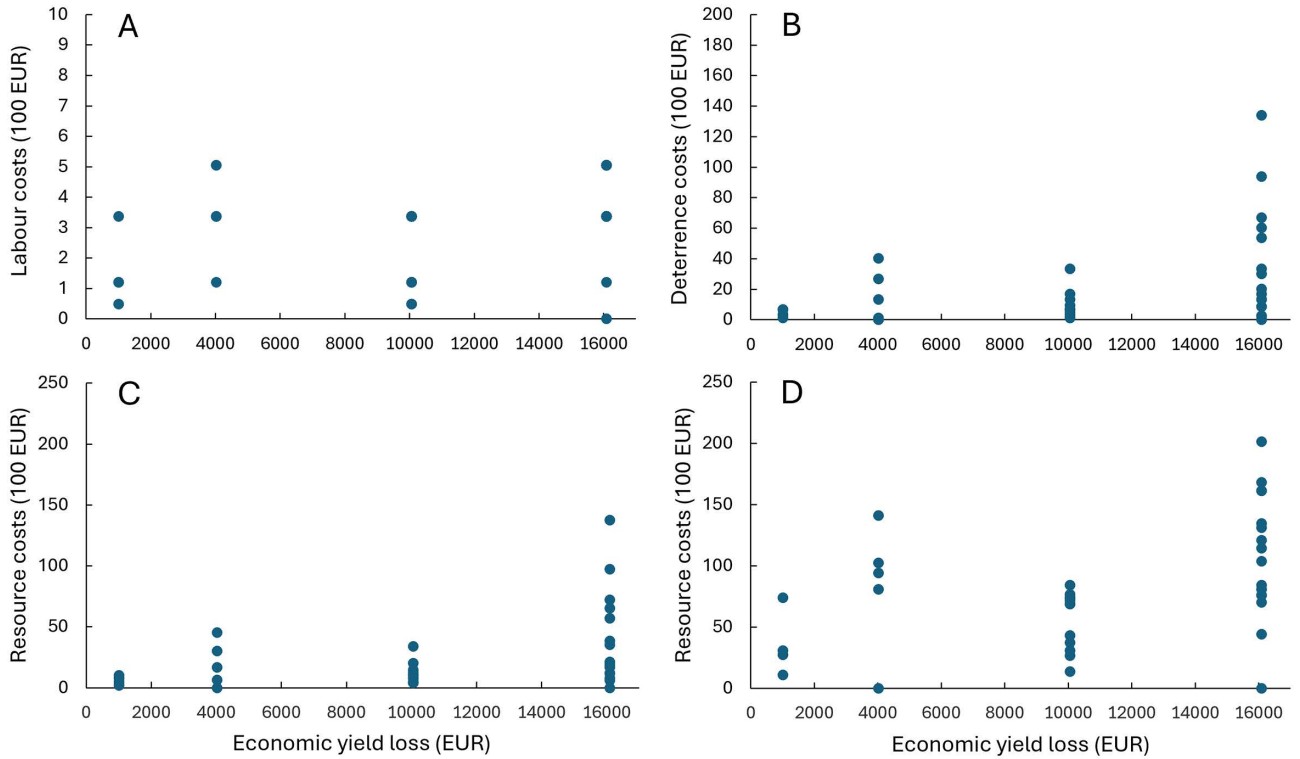

**Fig 5. Assessed economic costs of goose handling by farmers in relation to perceived annual yield loss.** Costs for handling the goose problems are expressed in time consumption per week converted into labour costs (EURO) (A), costs of deterrence equipment and fuel for driving (B), total resource costs (labour costs + equipment + fuel costs) for one week of mitigation (C), and a scenario for 20 weeks of mitigation efforts (D). In the scenario for 20 weeks of mitigation, costs for deterrentsand fuel are kept fixed, while time expenditure increase with length of the mitigation period.

**Table 5. Hunters availability to perform derogation shooting of barnacle geese.**

| Hunters availability | Derogation is a part of the hunting lease agreement | Derogation is not a part of the hunting lease agreement |
|---|---|---|
| Coming rarely + not always | 1 (4.5) | 9 (5.5) |
| Coming often | 12 (8.5) | 6 (9.5) |

Distribution of observed and expected (under equal distribution) responses regarding whether the respondents have hunting lease agreements where derogation shooting is included as part of the agreement with the hunters, or if derogation is not part of the hunting lease agreement, and how the respondents perceive the hunters' availability to come and carry out barnacle goose derogaton shooting when needed (rarely + not always vs. often).

To minimize the energy expenditure for flying, barnacle geese often seek food within a four km radius of their roosting sites on water bodies [23], although this distance can increase in certain cases if the gain outweighs the effort [27]. The geese prioritize areas with low vegetation, high energy content, and low fiber content. To minimize energy consumption, geese generally try to avoid hazardous areas, to reduce risk of predation. Therefore, they prefer safe large fields over 10 hectares, with good visibility and no human disturbance [23]. Barnacle geese primarily forage during daylight hours but may also forage at night, particularly if there is moonlight [28]. The strategy is to optimize energy intake during the short period of daylight available in the winter season.

With the above energetic demands and behaviour of barnacle geese in mind, the large wintergreen and coastal agricultural areas in Denmark are highly attractive foraging areas during the winter period, which increases the risk of crop damage and thus presents challenges for many farmers. In theory, the challenges can be solved by making the fields unattractive to geese, e.g., by planting crops they cannot/will not eat, reducing the size of fields with hedgerows and shrubs, or implementing continuous disturbance. In practice, these options may not align with modern agricultural practices, and farmers are therefore seeking more cost-effective solutions. Efficient deterrence of geese is thus about whether it is possible to find deterrents that are sufficiently cost-effective.

Situations can arise when geese go into energetic deficit, such as during continuous disturbance or periods of harsh winter weather. This may lead to more risk-prone behaviour in geese, where even the best deterrent may not have the expected effect. Therefore, it is impossible to determine the effectiveness of a deterrent in all situations.

## Active deterrents

According to the survey, active deterrents have the largest area effect. It is logical that using a vehicle can clear a larger area than, for example, a scarecrow, but when comparing it to the duration effect for each deterrent, it becomes clear that active deterrents have the shortest duration of all deterrents. The result is that active deterrents require continuous effort and thus may not be a cost-effective choice for deterring geese, especially if the farmer's fields are spread over a large area. Several respondents have reported that they often experience geese returning 10–30 minutes after being scared off by a vehicle or on foot. This aligns with what Simonsen et al. [29] found in an experimental study from Norway, where active scaring (going into the field) required scaring five times a day in the same field to reduce the use by pink-footed geese *Anser brachyrhynchus* in the spring. A study from England with scaring of brent geese *Branta bernicla* from wheat and rapeseed during winter showed that it was almost possible to keep the geese away using active deterrence (human+ATV). The study covered 100 hectares, and one ATV patrolled the fields continuously throughout the day, six days a week. An economic analysis showed that the effort cost 33–52 EURO per hectare, but when compared to acoustic and visual methods, which are cheaper to deploy, the active effort was still economically advantageous due to reduced yield loss caused by geese compared to other methods [30]. Several studies have tested active deterrents in various situations, and all find the same tendency: geese can be scared off with active deterrents, but to keep the geese away from the fields, it requires continuous effort and may therefore risks not being cost-effective [6–9]. To optimize the effect of active deterrence, it seems that provisioning alternative undisturbed foraging sites can be a positive factor [23,30,31].

Why active deterrents do not have a longer duration effect cannot be determined from this study, but we know from a study with GPS-tagged barnacle geese in the Guldborgsund Municipality that deploying active scaring, only a small proportion of the geese in the entire area experienced the disturbance. This resulted in new geese appearing shortly after the deterrence action, as they had not experienced the disturbance and could safely utilize the food resource [23].

From our interview survey we have no evidence that combining active deterrents with other deterrents can provide a longer duration effect.

## Passive deterrents

In contrast to active deterrents, passive deterrents have a smaller area effect, where a flag or bag on a stick is estimated to cover about one hectare. Agricultural machines placed in the field, hawk kites, and 'hanging human figures' have area effects up to 10 hectares. The median duration effect for passive deterrents is four to six days, except for 'hanging human figure', which is reported to have a duration effect of more than 21 days, however, with a small experience base. By deploying the four passive deterrents scarecrows, flags/strips/bags, silhouette figures, and agricultural equipment significantly longer duration effect can be achieved compared to the three active deterrents: walking person, car/ATV, dog off-leash, and the auditory deterrent of scare shots. A study with deterrence of barnacle geese in Guldborgsund Municipality found that with passive deterrents, the geese returned after almost eight days on average. The return time for passive

deterrents was significantly longer than for active and auditory deterrents (J. Madsen, Aarhus University, in prep.). With our study and the experimental work in Guldborgsund Municipality, it appears that passive deterrents can achieve longer duration effects than both active deterrents and scare shots. However, with passive deterrents, once the duration effect has been 'used up' because geese habituate, new deterrents must be found (according to several respondents, moving scaring devices does not have a real effect), unlike with active deterrents and scare shots, where continuous effort maintains effectiveness throughout the season. A study from North America found that by placing 2.5 white flags per hectare on rye and wheat fields, farmers could keep snow geese *Chen caerulescens* away from the fields for at least 5.5 weeks [32]. The study did not observe habituation, but it would be expected if the experiment had lasted longer.

If a longer lasting effect of passive deterrents is to be expected, the survey suggests that at least one deterrent per hectare should be set up. We have reports from farmers who make significant efforts with passive deterrents and experience good results, placing sticks with strips or big bags at high densities in the most heavily impacted fields. One diligent farmer sets up sticks with a density of 15–20 per hectare on heavily impacted fields, which in this specific case leads to a complete abandonment by barnacle geese. This is a labour-intensive solution, but for this farmer, it is a satisfying solution.

### Auditory deterrents

The auditory deterrents in this study include scare shots, gas cannons, and scare sounds, such as bird-alerts. They have a relatively large area effect of between 11–20 hectares, which is in accordance with Bishop et al. [5], who describe that to protect crops from waterfowl, one gas cannon is placed per 20 hectares. If the two stationary auditory deterrents (bird-alert and gas cannons) are maintained and moved around, the duration effect in our study is estimated to last up to 21 days before the geese habituate. By moving the gas cannons every five days and setting them to fire at variable intervals, habituation can be delayed [5]. In connection with deterring birds from airport areas, Davis [33] recommends moving the gas guns every two to three days and finds them most effective against huntable species. With these auditory deterrents, good results can be achieved, but there are several drawbacks. Most importantly, auditory deterrents risk causing disturbance in the surroundings, not only to other fauna but also people in the local area. Exposure to acute noise over 80 dB activates the nervous system and leads to hormonal stress responses in both humans and other mammals [34]. Additionally, these deterrents require continuous maintenance, recharging of batteries, and replacement of gas, etc.

Scare shots (possibly with pyrotechnics) have a relatively short duration effect of 1–3 days. However, with this method, it is expected that the effect can be maintained throughout the season through repeated interventions. A group of farmers have reported good results with scare shots, where, in collaboration with the local hunting association, they consistently patrol the fields several times a day throughout the season and use scare shots for deterrence with satisfactory success. Furthermore, the effect can be optimized if the geese have alternative foraging sites in the area [5].

### Combined techniques

Dead geese in the field is a method that uses the geese's real fear of white-tailed eagles, so there is no risk of habituation as with many other methods that imitate danger. Many respondents see this as a good solution since it is passive and therefore 'takes care of itself'. However, it requires active effort to take down geese for this purpose, which again requires a permit and compliance with the conditions of the permit. Furthermore, it can be debated whether laying out dead game is ethically defensible in hunting, as stated in the Danish hunting ethics rules: "Dead game must, whenever possible, be used for food or fur, as a trophy, or in some other reasonable way. This also applies to game killed under the regulations in the ordinance on wildlife damage" [35]. The method of laying out dead geese also has a short duration effect of 3–4 days, which is probably because the dead geese are naturally eaten/removed by predators, but while it 'lasts', a dead goose can protect around 10 ha due to the presence of a white-tailed eagle.

Drones are an example of a potentially smart deterrent tool, but one that currently cannot meet the challenges satisfactorily. The special potential of drones lies in the possibility that they may be able to fly autonomously in the future, thereby

saving resources by reducing labour costs. Currently, drone flying is heavily regulated, both by the EU's drone regulation (2019/947 and 2019/945) and Danish legislation (Regulation on supplementary provisions to implement regulation (EU) 2019/947) on the rules and procedures for the operation of unmanned aircraft. These provisions limit the possibilities of using drones as a cost-effective deterrent, and thus it is not yet a feasible solution. A recent study by Månsson et al. (2024) from Sweden on the use of drones for deterring geese mentions several advantages of drones as a deterrent, such as long range, potential for quick response time, gentleness towards crops, options for various expressions, sounds, and lights, and potential for automatic use. However, similar to our study, they also note that the deterrent effect of drones quickly fades, as no effect is detected 24 hours after the intervention, i.e., comparable to active deterrents, which have a relatively large area effect but a short duration.

Derogation shooting of barnacle geese is a deterrent action that can be applied after prior approval, when other methods have proven insufficient. Respondents report positive effects on the duration effect when hunters manage to take down individuals. Several respondents report that with larger shooting actions, where more hunters are involved over larger continuous areas and take down several barnacle geese, this leads to longer periods of goose absence on the affected fields. It is expected that the longer effect is partly achieved because many more geese are exposed to the deterrence action starting at sunrise when the geese fly onto the field, and possibly for several hours thereafter.

Månsson [36] tested the effect of lethal deterrence on summering greylag geese in Sweden and showed a significant decrease in the number of geese on the fields where deterrence occurred. The effect could be observed for at least three days, but the observations were not continued for longer. In Guldborgsund Municipality, it has been experimentally shown that derogation shooting of barnacle geese has a duration effect of 6–8 days on average, approximately twice as long as active and auditory deterrents (J. Madsen, Aarhus University, in prep.). A study from Islay, Scotland, found that low-intensity derogation shooting of barnacle geese had little effect on redistributing geese from vulnerable fields to nearby refuges. Therefore, it is suggested that the intensity of shooting is increased at vulnerable sites to maximize the flock's perception of risk [37]. The shooting effort can also increase the effectiveness of other non-lethal deterrents, as many of these attempt to instill fear of lethal deterrence, e.g., gas cannons, scarecrows, scare shots, etc. [36]. One respondent reports using this by having the geese associate certain passive deterrents with the fear of lethal regulation. This is in line with reports from several respondents who say that the geese flee from selected vehicles used for daily active deterrence, probably because they have learned to associate the vehicle with danger.

### Assessment of effectiveness across deterrence methods

The ideal for farmers is to achieve both a large area effect and long duration. The overview of effectiveness of combined area and duration effects (Fig 4) points out that derogation shooting, scare sounds, and gas cannons are in the highest quadrant of effectiveness. However, it should be considered that these types of deterrents require adjustments to maintain long-term effects. Most passive deterrents fall into the lowest quadrant, except for the 'hanged human figure', which appears more promising in terms of duration (but more experience should be gathered). Meanwhile, most active deterrents fall into the quadrant with high area effect and low duration. The study shows that users can increase the effectiveness of passive deterrents by using higher densities, which will affect both duration and area effect. Additionally, there are user experiences and experimental evidence that the effectiveness of active deterrents can be improved by increasing the frequency of their use, which will positively affect the duration.

To increase effectiveness, it will require greater effort from users, both in terms of time and material investments. Thus, there is a time and economic trade-off relative to the perceived extent of the problem.

### The extent of the problem

This study shows that many farmers in the coastal areas of Denmark experience overwintering barnacle geese as a burden, which is also reflected by the widespread applications for permits to shoot barnacle geese under derogation

[38]. Foraging barnacle geese on agricultural land are reported to pose a burden, leading to yield losses, significant time investments in deterrence efforts, costs associated with deterrent measures, and frustrations over the problem which 'does not go away' as well as lack of tools to remedy the problem. Among the respondents, although not representative of the farming community as a whole, the average reported yield loss is 10,800 EURO per year and resource costs 8,000 EURO (rounded to nearest 100). For resource costs we here use a mitigation period of 20 weeks, which seems to be the most prevalent situation.

The actual, quantitatively measured yield losses and farmers' perceived yield losses do not necessarily align [14,39]. We have based our analysis on farmers' assessments since their perception of the issue ultimately determines their willingness to invest in deterrence. To compare, we have direct quantitative measurements of the damage caused by barnacle geese to winter wheat in Guldborgsund Municipality [14]. This study found up to 6% yield loss after winter and spring foraging. Foraging in May can cause a yield loss of up to 10% (K. K. Clausen & J. Madsen, unpubl.). With a current sales price for winter wheat (average of 2023 and 2024) of 0.214 EURO per kg and an average yield of 6.5 tons per ha in Denmark [40], 6% loss represents 83.62 EURO per ha. The average agricultural area per property in our study is 528 ha, of which around one third, equivalent to 174 ha, can be assumed to be covered by winter cereal fields [23]. In a situation where all winter cereal fields are used by geese, this would then result in an average total yield loss of 14,600 EURO. According to the field-based assessments, not all fields are likely to be used [40], and the damage may be negligible in some fields. Nevertheless, the estimates based on the reports by farmers and the upscaled field experiments are in the same orders of magnitude, which gives confidence that the estimates based on the report by farmers are not seriously biased, though probably in the high end because not all fields are likely to be used by geese during spring when the damage is most likely to occur.

To put the economic costs (yield loss plus resource costs) into a farming economy perspective, we can compare these to the average operating profit for Danish full-time agriculture with crop production. On average for 2023 and 2024, the operating profit is reported to be 92,269 EURO and, after owner remuneration, 30,083 EURO [41]. Hence, the reported average goose yield loss and resource costs (18,800 EURO), represent between 20 and 63% of the operating profits, depending on whether the goose induced costs have to cover the salary of the owner or not. Even if the farms selected for the study may be at the high end of the size of the average farm used in the statistics and therefore have a higher operating profit, the calculation shows that the amount of the profit lost to cover goose damage, time and material is sizeable for the farmer's economy.

We find significant variation in how much is invested in deterrence, despite substantial reported yield losses. This may suggest that some farmers either tolerate the problem or have given up on deterrence, regarding it to be ineffective or too time-consuming. This is reflected in the responses, where most respondents believe their overall deterrence activities solve their problems "to some extent," while 13% consider the issue of barnacle geese to be an "overwhelming burden." The lack of investment may also stem from farmers' unawareness that more effective deterrence methods could be deployed, either within their own property or in cooperation with neighbours and hunters.

## Hunting lease agreements

To reduce the time spent on deterrence and increase effectiveness, most of the farmers collaborate with hunters who conduct barnacle goose derogation shooting on their behalf, highlighting the crucial role hunters play in mitigating the problems with geese. On the one hand, for the hunters, derogation shooting gives more goose shooting opportunities. On the other hand, a challenge arises as hunting is a recreational activity for these hunters, and effective derogation shooting of barnacle geese often demands more effort than their interest can sustain. Furthermore, the need for shooting action may arise at times when hunters are at work or otherwise unavailable. This mismatch creates a conflict, as the responsibility of shooting barnacle geese begins to resemble an obligation, surpassing the voluntary motivation inherent in a hobby.

The study indicates that respondents are more satisfied with their hunters when they have entered into a hunting lease agreement where the derogation shooting of barnacle geese is explicitly included in the contract. Through individual reports, we are aware of several such agreements in which the hunting lease fee has been reduced, or alternative hunting opportunities have been granted, in exchange for hunters committing to perform derogation shooting barnacle geese. This type of specific agreement, in which clear expectations are established before the season, appears to foster successful collaboration.

Additionally, we are aware of two areas in Denmark where farmers have joined forces to regulate barnacle geese in cooperation with hunters. In one area, members of a local hunting association take turns patrolling vulnerable crops, with farmers compensating them for travel expenses. In another location, joint derogation shooting events are conducted, where the farmers' hunting tenants carry out the shooting, coordinated by one of the farmers. Both arrangements seem to be effective. A common feature of these approaches is that they require cooperation across property boundaries and between hunters and farmers.

## Accommodation fields

The concept of establishing accomodation fields to alleviate pressure on more vulnerable crops has become well known and widely adopted among farmers in the Guldborgsund Municipality project area. However, it does not yet appear to be a common strategy among farmers in other regions. Around half of the respondents indicate that they use accommodation areas, and the majority do so deliberately to mitigate issues caused by foraging barnacle geese. When asked whether they believe these areas have the desired effect, 62% responded affirmatively. The perception that accommodation fields provide a mitigating effect aligns with findings from other studies [9,30,31,42].

When geese are scared away from a field, they are often merely displaced within the local area, leading to increased energy expenditure for the birds [43] and requiring continuous scaring efforts from farmers. This means the problem is relocated rather than resolved. The rationale behind accomodation fields is to provide a peaceful foraging area for geese, where hunting and deterrence are not practiced. This study finds that most respondents who have accomodation fields ensure these areas remain hunting-free.

Previous studies from Denmark have shown that both leftover sugar beet fields and maize stubble fields can serve as attractive foraging areas for barnacle geese, thereby alleviating pressure on vulnerable winter crops in autumn [23,27]. Furthermore, it has been demonstrated that in spring, barnacle geese prefer grasslands, particularly saltmarshes, provided that the vegetation is short [23]. However, many saltmarshes have become overgrown and are no longer attractive to geese, suggesting that there is untapped potential in the landscape for accomodation areas [44].

The concept of accomodation fields is an ethically sound and holistic solution that aims to resolve the issue rather than merely shift it elsewhere. To be effective, attractive foraging areas must be available throughout the winter, either with crop residues and/or various types of grasslands. Effectiveness of a local goose management scheme is further enhanced when accomodation fields are combined with intensive disturbance in vulnerable areas, creating a significant perceived risk for geese – such as through intensive derogation shooting combined with passive deterrents [37]. However, in situations where such local management schemes are dependent on public compensation for yield losses and subsidies to allow geese to forage in agricultural fields used as accommodation areas, scaring may not be cost-effective [45]. In the Danish setting where no public money is provided in support of local goose management, a voluntary organization of 'push-pull' between scaring and accommodation areas is clearly cost-effective because the majority of herbivorous waterfowl feeding on vulnerable farmland can be concentrated in accommodation areas set up without costs, at least for parts of the wintering season [23].

## Methodology and uncertainties

This study has been conducted through a survey, where we attempt to draw general conclusions based on a sample of 54 respondents. Naturally, this is a small sample size, but several of our findings have proven to be highly significant.

We have compensated for the relatively small sample size by improving the quality of our interviews conducted. This is achieved by conducting them face-to-face, which ensures higher response retention and completion rates while also allowing for a better understanding of the questions. Additionally, this method provides supplementary insights that would not have been captured through written or online surveys. As a result, numerous anecdotal reports have been gathered, which prove useful in understanding effective deterrence strategies for geese.

When comparing our results with existing scientific studies, we find a reasonable level of agreement, reinforcing our ability to draw general conclusions based on this user survey. The study should not be confused with natural scientific research. Instead, it is based on perceived and self-reported effects of deterrent methods and economic burdens, making it inherently less exact. Nevertheless, given that we have obtained highly significant results validated by existing literature and that economic reports of damage are at least to some degree in line with upscaled quantitative field assessments, we believe it is possible to derive meaningful conclusions.

The advantage of using a user survey is that we have been able to obtain multidimensional responses regarding the effectiveness of various deterrent measures, something that would have required extensive large-scale experiments to clarify. While we have not achieved the precision and control that a scientific block design would provide, we have instead gathered numerous responses that can be considered replicates, as the study primarily focuses on a single species and a single crop type, namely winter wheat.

## Conclusions

The survey has provided an overview of user experience regarding the effectiveness of deterrence measures used to mitigate crop damage caused by barnacle geese. Since barnacle geese have proven to be highly adaptable in their use of agricultural landscapes, similar to other goose species, it is likely that this knowledge can largely be applied to other species as well. As barnacle geese are increasingly subjected to derogation shooting [38], their behaviour has probably become more comparable to that of huntable species.

The most effective deterrence measures are characterized by long-lasting effects and extensive area coverage. In general, passive deterrence methods remain effective for about 4–6 days (until the geese become habituated) but have a relatively small area effect. In contrast, active measures generally have a higher area effect but a shorter duration. The greatest area coverage and longest-lasting effect are achieved through gas cannons and combined methods such as scare sounds and derogation shooting. However, the use of gas cannons can be problematic due to noise disturbances. More intensive active deterrence or a significantly increased density of passive deterrents can improve effectiveness. Both approaches require a greater investment in time and deterrence resources from the users.

Derogation shooting performed by hunting leaseholders requires that hunters are willing and available to respond on short notice when needed by the farmer. Establishing hunting lease agreements that include clear expectations regarding the hunters' role in mitigating goose damage and performing derogation shooting appears to enhance cooperation and problem-solving. Local collaboration between farmers and hunters seems to further increase effectiveness and satisfaction.

To prevent geese from simply being chased back and forth between vulnerable crop fields due to deterrence measures, the establishment of accomodation fields with attractive food sources, such as spilt grain or maize, sugar beet residues, or cultivated grass, can be beneficial. Here, geese should be allowed to forage undisturbed. This approach also takes greater ethical considerations into account by ensuring that the geese can fulfill their ecological and behavioral needs.

There is a relationship between farmers estimated financial losses due to goose-related crop damage and their resource expenditures in terms of time and deterrence measures, but with marked variation. A lack of investment may indicate that farmers either find the problem manageable or have given up on deterrent efforts. It may also be due to

a lack of awareness that deterrence efforts can be intensified through more effective methods, either within their own property or in cooperation with neighbouring farmers and hunters. Given that yield losses caused by geese and resources spent on scaring geese may have a sizeable impact on the operation profits of farmers, there is a financial incentive of entering cooperation and establishing joint accommodation areas.

Having taken a transdisciplinary approach to the assessment of effective deterrence, built on users' own knowledge and perceptions as well as a dialogue about the knowledge needs by the authorities to improve goose management, we trust that the study and its communication can lead to an enhanced knowledge uptake and social learning [46]. This may provide a basis for a change in the farmers' willingness to engage more actively in solving the local goose conflict [47]. This approach can result in a broader basis for advising on goose management for the benefit of both farmers facing these challenges and authorities providing guidance and ensuring compliance with national and international legislation. Analysing the utility of collaboration, our study feeds into ongoing attempts in Denmark to set up locally anchored co-management of mobile wildlife such as geese and deer as a way forward to solve human-wildlife conflicts.

## Acknowledgments

We extend our sincere thanks to the respondents for participating in the interviews. The Agency for Green Transition and Aquatic Environment and the Nature Agency for introducing us to the issue, and special thanks to Jesper Pedersen (Nature Agency – Storstrøm) for assistance in answering technical questions. Thanks to Hans Peter Hansen and James Henty Williams for their support on social science methodology and Peter Mikkelsen for organising the database for analysis.

## Author contributions

**Conceptualization:** Sandie Lohse Sørensen, Jesper Madsen.

**Data curation:** Sandie Lohse Sørensen, Thorsten Johannes Skovbjerg Balsby.

**Formal analysis:** Sandie Lohse Sørensen, Jesper Madsen, Thorsten Johannes Skovbjerg Balsby.

**Funding acquisition:** Jesper Madsen.

**Investigation:** Sandie Lohse Sørensen, Jesper Madsen, Thorsten Johannes Skovbjerg Balsby.

**Methodology:** Sandie Lohse Sørensen, Jesper Madsen, Thorsten Johannes Skovbjerg Balsby.

**Project administration:** Jesper Madsen.

**Resources:** Jesper Madsen.

**Software:** Thorsten Johannes Skovbjerg Balsby.

**Supervision:** Jesper Madsen.

**Validation:** Sandie Lohse Sørensen, Jesper Madsen, Thorsten Johannes Skovbjerg Balsby.

**Visualization:** Sandie Lohse Sørensen, Jesper Madsen.

**Writing – original draft:** Sandie Lohse Sørensen, Jesper Madsen, Thorsten Johannes Skovbjerg Balsby.

**Writing – review & editing:** Sandie Lohse Sørensen, Jesper Madsen, Thorsten Johannes Skovbjerg Balsby.

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
