## [Decision Letter · Decision Letter 0]

3 Nov 2025

PLOS ONE

Dear Dr. Madsen,

Thank you for submitting your manuscript to PLOS ONE. After careful consideration, we feel that it has merit but does not fully meet PLOS ONE’s publication criteria as it currently stands. Therefore, we invite you to submit a revised version of the manuscript that addresses the points raised during the review process.

This work is well prepared and well presented, and it follows the scope of PLos one, and principle of inclusion. Reviewer 1 is very positive, but also points at several presentation and conceptual improvements that are needed in order to accept your work for publication.

We look forward to receiving your revised manuscript.

Kind regards,

Benedicte Riber Albrectsen

Academic Editor

PLOS ONE

Journal Requirements:

“The project was primarily funded by a grant from the Danish Agency for Green Transition and Aquatic En-vironment to Jesper Madsen (www.sgav.dk)(Jagttegnsmidlerne 2023; ref. 120362), with support from a grant by the 15 June Foundation to Jesper Madsen (www.15junifonden.dk) (project Samforvaltning i praksis – forsøg med bramgæs i Guldborgsund Kommune; ref. 2022-022).”

4. Thank you for uploading your study's underlying data set. Unfortunately, the repository you have noted in your Data Availability statement does not qualify as an acceptable data repository according to PLOS's standards.

Additional Editor Comments:

This work is well prepared and presented, and follow the scope of inclusion of PLos one. Reviewer 1 is very positive, but also points at several presentation and conceptual improvements that are needed in order to accept your work for publication.

Reviewer's Responses to Questions

**Comments to the Author**

1. Is the manuscript technically sound, and do the data support the conclusions?

Reviewer #1: Yes

2. Has the statistical analysis been performed appropriately and rigorously?

Reviewer #1: Yes

3. Have the authors made all data underlying the findings in their manuscript fully available?

Reviewer #1: Yes

4. Is the manuscript presented in an intelligible fashion and written in standard English?

Reviewer #1: Yes

Reviewer #1: Line 352: "go out into field" not in table. I assume this should read: "walking person". Likewise, Figures 2 & 3 have listed "Walking man". This should be consistent with the other references to this hazing method.

Figure 2/Line 373-375: Claims significance but does not state p-value in text. Horizontal lines indicating significance in the figure are distracting. Consider using error bars or indicate significance through: A, B, BC, C above the bar graphs.

Overall, throughout text, be sure to use past-tense and fix grammatical errors, especially the use of ", and" at the end of sentences.

Line 394/Table 3/Figure 3: Throughout the publication, choose your desired p-value and stick with it. In the text/table 3, it appears that the p-value is referring to <0.001 suggesting. Then in figure 3, it appears that the error bars are set at maybe p-value of 0.05 (scare shots/walking man/driving car/atv all show similar significance, while in the table 3 they depict differences) . . . I think 0.05 is perfectly fine for these data but be consistent throughout.

Table 4: Needs more explanation in the text and table (and format DF/p-value/F to be consistent at least). After reading numerous times, I am still not sure what table 4 is showing me. Season number vs hectare numbers that are of little value to the reader.

Table 5: Similarly, needs more explanation in text and formatting is too bad to interpret.

Figure 5: Needs more explanation in text and needs a more self-explanatory figure overall. (This data is covered exceptionally well in the discussion. Potentially remove the figure altogether and leave those data review for the text.)

Table 6: Consider moving the "Hunters come ... " portion of the first column text to the top row and change the text to "Hunters available"

I was very impressed with the Discussion. The authors have a strong understanding of the issue. Likewise, I liked figures 1, 3, 4 and tables 2 and 3. Those were great visuals. Overall, well done!

**Do you want your identity to be public for this peer review?** For information about this choice, including consent withdrawal, please see our Privacy Policy

Reviewer #1: No

---

## [Author Response · Author response to Decision Letter 1]

22 Dec 2025

I have a problem opening the word files with the Reply to Reviewers while working in the PLOS One system. I refer to the file addressing the comments

---

## [Editor Report · Decision Letter 1]

13 Jan 2026

An analysis of farmers' experiences with deterrence methods and investment in mitigation of agricultural crop damage caused by geese

PONE-D-25-42741R1

Dear Dr. Madsen,

We’re pleased to inform you that your manuscript has been judged scientifically suitable for publication and will be formally accepted for publication once it meets all outstanding technical requirements.

Kind regards,

Benedicte Riber Albrectsen

Academic Editor

PLOS One

Additional Editor Comments (optional):

The authors have responded adequately to every question raised by the reviewer, who was already impressed with their work.
---

## [Editor Report · Acceptance letter]

PONE-D-25-42741R1

PLOS One

Dear Dr. Madsen,

I'm pleased to inform you that your manuscript has been deemed suitable for publication in PLOS One. Congratulations! Your manuscript is now being handed over to our production team.

Kind regards,

on behalf of

Dr. Benedicte Riber Albrectsen

Academic Editor

PLOS One